# Tailoring of Dissimilar Friction Stir Lap Welding of Aluminum and Titanium

**DOI:** 10.3390/ma15238418

**Published:** 2022-11-26

**Authors:** Alexander Kalinenko, Pavel Dolzhenko, Yulia Borisova, Sergey Malopheyev, Sergey Mironov, Rustam Kaibyshev

**Affiliations:** Laboratory of Mechanical Properties of Nanostructured and Heat Resistant Materials, Belgorod National Research University, 85 Pobeda-Street, Belgorod 308015, Russia

**Keywords:** dissimilar friction stir lap welding, 6013 aluminum alloy, Ti–6Al–4V alloy, intermetallic compound, grain structure, lap-shear test

## Abstract

An approach was proposed to optimize dissimilar friction stir lap welding of aluminum and titanium alloys. The basic concept of the new technique included (i) the plunging of the welding tool solely into the aluminum part (i.e., no direct contact with the titanium side) and (ii) the welding at a relatively high-heat input condition. It was shown that sound welds could be readily produced using an ordinary cost-effective tool, with no tool abrasion and no dispersion of harmful titanium fragments within the aluminum side. Moreover, the intermetallic layer was found to be as narrow as ~0.1 µm, thus giving rise to excellent bond strength between aluminum and titanium. On the other hand, several important shortcomings were also revealed. First of all, the high-heat input condition provided significant microstructural changes in the aluminum part, thereby resulting in essential material softening. Furthermore, the new approach was not feasible in the case of highly alloyed aluminum alloys due to the relatively low rate of self-diffusion in these materials. An essential issue was also a comparatively narrow processing window.

## 1. Introduction

The rapidly growing engineering requirements in the transportation industry necessitate the implementation of hybrid structures consisting of a combination of several different materials. Among such combinations, lightweight aluminum–titanium structures represent a particular interest due to both economic and technical benefits. An essential problem, however, is the incompatibility of the thermal properties of these two metals, which significantly restricts the application of conventional fusion welding. Hence, innovative friction stir welding (FSW) is often considered as a possible candidate for the dissimilar joining of aluminum and titanium [1,2,3]. Being a solid-state technique, FSW avoids the formation of an undesirable solidification microstructure and, thus, enables the sound joining of various dissimilar materials.

Extensive research over the last two decades has conclusively demonstrated the feasibility of FSW for successive joining of aluminum and titanium alloys [1,2,3]. On the other hand, three important problems have been revealed.

One of the critical issues is the abrasion of the welding tool, which typically occurs during FSW of titanium alloys, e.g., [4]. To diminish this undesirable effect, high-cost tungsten-based tools are frequently used for dissimilar welding of aluminum and titanium [2,3].

The formation of intermetallic compounds at the aluminum–titanium interface is also of great concern [1,2,3]. Usually, intermetallic TiAl_3_ has been reported to form [2,3,5,6,7,8,9,10,11,12,13,14,15,16,17,18,19], presumably due to the comparatively low free energy of the intermetallic reaction in this case [20]. However, the details of this process are unclear. In some cases, TiAl [3,6,7,9,10,15] and Ti_3_Al [2,3,6,21] phases have also been found. As shown by Choi et al. [6], the development of TiAl and Ti_3_Al intermetallics is associated with a diffusion penetration of aluminum through the early-formed TiAl_3_ phase. Hence, the formation of TiAl and Ti_3_Al compounds occurs sequentially, one by one; moreover, the probability of these two processes is directly linked to FSW heat input. Specifically, it was shown that the increase in the tool rotation rate promotes the nucleation of the TiAl phase first and then the Ti_3_Al phase [6].

Due to their natural brittleness, all these intermetallics typically promote cracking and, thus, lead to the premature failure of dissimilar weldments [6,10,12,17,22,23,24,25,26]. It is well-accepted that the mechanical performance of aluminum–titanium dissimilar welds is mainly governed by the thickness and spatial distribution (continuous or discontinuous) of intermetallic compounds, e.g., [1]. The thickness is directly related to the weld heat input [1,5,13,27], while the spatial distribution is presumably determined by the character of material flow during FSW. Depending on particular FSW conditions, the thickness of the intermetallic layer has been reported to vary from 20 nm [6] to 2 µm [28]. It is believed that the critical thickness is 5 µm [3], while a thickness of less than 1 µm is considered acceptable [1]. It is important to emphasize that the growth of the intermetallic layer is usually attributed to the diffusion-assisted penetration of aluminum into titanium [3,20].

A significant problem is also the dispersion of titanium fragments in the aluminum part, which also typically occurs during dissimilar FSW of aluminum and titanium [5,6,7,8,14,15,18,23,25,29,30,31,32,33]. This effect also results in micro-cracking and, thus, finally leads to the early fracturing of the welded joints [1,6,7,29].

To minimize the detrimental influence of the above shortcomings, an appropriate optimization of FSW technology is necessary. In the *lap-welding* configuration, this is achieved by the placing of an aluminum plate on the upper part of the welding joint, so the welding tool is mainly plunged into the relatively soft material [2,3]. This simple approach reduces tool abrasion while also localizing the spatial distribution of titanium fragments in a relatively small area [1,3,10,22]. Moreover, the minimizing of the interaction between the welding tool and titanium lowers the FSW heat input [2] and, thus, shrinks the intermetallic layer [1,5,6,13,15,22,32,34,35].

Ideally, the exclusion of direct contact between the welding tool and titanium (i.e., the plunging of the tool solely into the aluminum part) should eliminate all the above problems completely. However, the joint efficiency of such welds has been found to be low [9].

This confusing result was likely due to the insufficient FSW heat input. Indeed, considering the lack of mechanical intermixing of the welded materials in this case, the welding mechanism should be virtually close to diffusion bounding (though the diffusion processes are enhanced by the large plastic deformation in aluminum). If so, the peak temperature and the duration of the FSW thermal cycle are of key importance.

Therefore, the present work was undertaken in an attempt to tailor the dissimilar FSW of aluminum and titanium alloys in the *lap-welding* configuration. In contrast to the previous works in this area, the proposed approach was based on two principal issues: (i) the zero penetration depth of the welding tool into the titanium part and (ii) the high FSW heat input. If successful, this technique should enable sound welding using the ordinary cost-effective FSW tool.

## 2. Materials and Methods

The baseline materials used in the present study included commercial AA6013 aluminum alloy and Ti–6Al–4V titanium alloy. The nominal chemical composition of AA6013 aluminum alloy is shown in Table 1. In order to investigate the possible influence of the important alloying elements in aluminum alloys (i.e., magnesium and silicon) on the feasibility of FSW, four additional experimental aluminum alloys were also utilized, as shown in Table 2. All aluminum alloys were produced by semi-continuous casting using laboratory equipment at Belgorod National Research University. The cast ingots were homogenized at 550 °C for 4 h and then cold-rolled to a final thickness of 2 mm (≈80% of total thickness reduction). Titanium alloy was received as 2 mm thick plates in the mill-annealed condition.

The aluminum and titanium plates were *lap-welded* using a commercial AccurStir 1004 FSW machine (Appendix A). In all cases, aluminum alloys were placed on the upper side of the lap joint, and the welding tool was plunged only into aluminum plates. Based on preliminary experiments, the distance between the probe tip of the plunged tool and the aluminum–titanium interface was kept at ≈50 µm. The ordinary welding tool was utilized for FSW. It was manufactured from tool steel and had a typical design, i.e., a concave-shaped shoulder (12.5 mm in diameter), and an M5 threaded cylindrical probe (1.9 mm in length). To provide the highest possible FSW temperature, the maximal tool rotation rate (allowable by the FSW machine) of 1100 rpm was used. In order to investigate the possible influence of the duration of the weld thermal cycle, a series of FSW trials were conducted at the welding rates of 0.5, 1, 2, 3, 4, 8, 12, 16, and 20 inches/min (12.7, 25.4, 50.8, 76.2, 101.6, 203.2, 304.8, 406.4, and 508 mm/min, respectively). In all cases, FSW was performed in the plunge-depth control mode, employing a tool tilting angle of 2.5° and a stainless backing plate. A typical convention for FSW geometry was adopted with WD, ND, and TD being the welding direction, normal direction, and transverse direction, respectively. In selected welds, the FSW thermal cycle was measured using the K-type thermocouples placed at the aluminum–titanium interface in close proximity to the stir zone. The schematic of the thermocouple placement is shown in Appendix A.

Microstructural observations were conducted using optical microscopy, scanning-electron microscopy (SEM), energy dispersive spectroscopy (EDS), and electron backscatter diffraction (EBSD). In all cases, microstructural samples were machined from the transverse cross section of welded joints (ND × TD plane). The final surface finish was obtained using conventional metallographic techniques with a final polishing step comprising 24 h vibratory polishing with colloidal silica suspension. The selected samples were additionally etched in Keller reagent. Optical examinations were carried out with an Olympus GX51 optical microscope using SIAMS 800 software. SEM, EDS, and EBSD observations were performed using an FEI Quanta 600 field emission gun scanning electron microscope equipped with TSL OIM^TM^ software and operated at an accelerated voltage of 20 kV. A 15-degree criterion was applied to differentiate low-angle boundaries (LABs) from high-angle boundaries (HABs) in EBSD maps.

To assist in interpretation of microstructure distribution, microhardness profiles were measured across the mid-thickness of the aluminum part of the welded joint. The Vickers microhardness data were collected using a Wolpert 402MVD microhardness tester by applying a load of 200 g, a dwell time of 10 s, and a step size of 0.5 mm.

The mechanical performance of welded joints was examined by lap-shear tests. The appropriate specimens were machined perpendicular to the WD and had a gauge section 35 mm in length and 6 mm in width. The specimens encompassed the entire width of the weldments (Appendix A) and, thus, included all the characteristic FSW zones, i.e., the stir zone, the thermo-mechanically affected zone, and the heat-affected zone. To achieve uniform thickness and eliminate surface defects, the face surfaces of the specimens were mechanically polished. The lap-shear tests to failure were performed using an Instron 5882 universal testing machine at ambient conditions and a nominal strain rate of 10^−3^ s^−1^. The distribution of local strain during lap-shear tests was measured by the digital image correlation technique using a commercial Vic-3D system. To ensure reliability of experimental results, three specimens were tested for each material condition.

## 3. Results and Discussion

### 3.1. Weldability

A series of welding trials have shown that the proposed approach is generally feasible for dissimilar welding of aluminum and titanium. Specifically, sound welds were obtained in the case of the commercial AA6013 alloy as well as in that of the experimental AA6013 + 2.0Si and AA6013 + 0.8Si + 1.0Mg alloys. A typical appearance of the welded joints is shown in Figure 1. On the other hand, FSW was not successful in the case of the highly alloyed aluminum alloys, i.e., AA6013 + 3.0Mg and AA6013 + 5.0Mg.

It is also important to emphasize that sound welding was only achieved at a welding speed of ≤76.2 mm/min. This observation emphasized the key idea of the present work on the principal significance of the weld heat input for the “zero-penetration-depth” welding. Given the nominal diffusion mechanism of joining in this case, the duration of the FSW thermal cycle should play a crucial role.

Using optical microscopy, the width of the welded surface was measured as a function of the alloying composition of aluminum alloys and the welding speed. A typical example is shown in Figure 2. Depending on the particular FSW condition, this width was found to vary from 3.14 mm to 6.58 mm (Table 3), thus being relatively close to the diameter of the tool probe (i.e., 5 mm). Quite expectably, the width of the welded surface decreased with the welding speed and the alloying content of aluminum alloys (Table 3).

Remarkably, no apparent tool abrasion was found.

### 3.2. FSW Thermal Cycle

The typical thermal cycles recorded during FSW are shown in Figure 3. In all cases, the peak welding temperature was close to 500 °C. No significant influence of the alloying composition of aluminum alloys on the temperature profile was found (Figure 3a).

Notably, given the liquidus temperature of Ti–6Al–4 of 1660 °C (or 1933 K), the peak FSW temperature (i.e., 773 K) represented only 773/1933 ≈ 0.4 of the homologous temperature of titanium alloy, thus being relatively low. Accordingly, no essential diffusion activity of titanium could be expected.

On the other hand, the duration of thermal exposure was fairly sensitive to the welding speed (Figure 3b). This observation agreed with a number of previous works, e.g., [36], and evidenced that the diffusion-driven processes were most pronounced at the lowest tool travel rate.

### 3.3. SEM–EDS Observations

In order to obtain insight into the welding mechanism, the interphase surface between aluminum and titanium parts was studied using SEM and EDS techniques. In all cases, observations were conducted at the center of the welded surface, as indicated in Figure 2b. The typical results are shown in Figure 4, Figure 5 and Figure 6.

In all FSW conditions studied in the present work, SEM observations revealed the formation of a thin (~0.1 µm) transition layer between aluminum alloys and Ti–6Al–4V. A typical example is indicated by the arrows in Figure 4a. Remarkably, the joint surface was not flat, thus evidencing the complex character of material flow in this area (Figure 4). Moreover, the refinement of beta-phase particles suggested plastic deformation in the upper part of the titanium side (Figure 4), despite the presumed “zero-penetration-depth” condition of FSW.

The typical results of the quantitative EDS analysis of the transition layer are shown in Figure 5. As expected, this layer represented an intermetallic compound. Remarkably, it had a composite structure consisting of TiAl_3_ and TiAl intermetallics. A similar effect has been reported in a number of previous works [3,6,7,9,10,15]. In the present study, the development of the TiAl phase was likely associated with the relatively low welding speed, which promoted the diffusion-driven penetration of aluminum through TiAl_3_. 

It is worth noting that the EDS measurements also revealed an increased concentration of silicon and manganese in both intermetallic phases (Figure 5). This observation was confirmed by the EDS elemental mapping, as shown in Figure 6.

Notably, EDS mapping revealed no titanium fragments within the aluminum part (a typical example is shown in Figure 6), thus evidencing that the welding tool did not come into direct contact with the titanium plate.

Surprisingly, no clear correlation between FSW conditions and the thickness or chemical composition of intermetallic compounds was found. In all FSW conditions examined in the present study, the intermetallic compound included TiAl_3_ and TiAl phases and the total thickness of the intermetallic layer was ~0.1 µm, as exemplified in Figure 4 and Figure 5, respectively.

### 3.4. Lap-Shear Tests

The typical results of lap-shear tests are shown in Figure 7. The entire set of the deformation diagrams is summarized in Appendix A. In most cases, the welded joints exhibited relatively high strength and significant elongation to failure. Remarkably, deformation diagrams typically showed serrations (Figure 7a). This suggested the occurrence of the Portevin–Le Chatelier effect and, thus, virtually implied the localization of the plastic deformation within the aluminum part of the welded joints.

In this context, it is worth noting that the welds produced at the lowest welding speed (i.e., the largest heat input) demonstrated a comparatively low strength (Figure 7a). The origin of this phenomenon is considered in Section 3.6.

Of particular importance was the observation that the failure usually occurred in the heat-affected zone of the aluminum part. This is in contrast to the typical deformation performance of dissimilar aluminum–titanium FSW joints, which frequently fail due to intermetallic compound cracking [6,10,12,17,22,23,24,25,26]. In the present study, such behavior was found only in the case of the welded joints of the magnesium-rich aluminum alloy AA6013 + 0.8Si + 1.0Mg produced at the highest welding speed of 76.2 mm/min (Figure 7b). Importantly, this welding condition showed the narrowest welded surface (Table 3).

In order to gain additional insight into the response of the welded joints to the lap-shear tests, the distribution of local strains was measured using a digital image correlation technique. The typical results are provided in Figure 8. These measurements conclusively demonstrated that the plastic strain was indeed localized within the aluminum part. Moreover, it was primarily concentrated beneath the edge of the tool shoulder (i.e., virtually in the heat-affected zone) on the advancing side of the weld. The possible origin of this behavior is considered in the following two sections.

### 3.5. Microhardness

It is reasonable to assume that the pronounced strain localization revealed during lap-shear tests originated from inhomogeneous microstructure distribution. To provide further insight into this issue, microhardness profiles were measured across the aluminum part of welded joints. The typical results are summarized in Figure 9. For clarity, the key tool dimensions are indicated in the profiles. To a first approximation, probe diameter delineates the stir zone, while shoulder diameter indicates the heat-affected zone.

In all cases, microhardness profiles exhibited the characteristic W-shape, with the lowest strength being measured in the heat-affected zone on the advancing side of welded joints. This observation was in excellent agreement with the digital image correlation measurements, which revealed strain localization (and subsequent failure) in this area (compare Figure 8 and Figure 9).

On the other hand, the stir zone material exhibited a comparatively high strength (Figure 9). Again, this result was consistent with the subtle plastic strain observed in this area during lap-shear tests (Figure 8).

Remarkably, the increase in the welding speed promoted a reduction in the softening effect in the heat-affected zone (Figure 9a). This result was also in line with the deformation behavior observed during lap-shear tests (Figure 7a).

Considering the perfect agreement between the microhardness profiles and the lap-shear tests, the former ones can be used for evaluation of the joint efficiency of welds. To a first approximation, this quantity could be expressed as a ratio of the lowest hardness in the heat-affected zone to the hardness of the initial material. From Figure 9b, it can be deduced that the joint efficiency ranged from ≈60% in the case of AA6013 + 0.8%Si + 1.0%Mg aluminum alloy to ≈70% for AA6013 or AA6013 + 2.0%Si aluminum alloys.

### 3.6. EBSD Measurements

To establish the relationship between microhardness variations and the underlying microstructure distribution, EBSD maps were taken from the initial material, the heat-affected zone, and the stir zone. The typical results are shown in Figure 10.

In the initial (cold-rolled) condition, the material exhibited a heavily elongated grain structure, which contained the dense LAB substructure (Figure 10a). This is a typical cold-rolled microstructure, which should provide an essential work-hardening effect.

In the heat-affected zone, the microstructure was dominated by the relatively coarse grains, which often had a nearly equiaxed shape and almost no LAB substructure (Figure 10b). The observed microstructural transformations suggested the occurrence of static recrystallization (and perhaps subsequent grain growth) in the heat-affected zone, as is normally expected in this area. This process should result in significant material softening, as indeed was observed in this microstructural region (Figure 9).

In the stir zone, significant grain refinement was observed (Figure 10c). This was also an expected result which is usually attributed to the continuous recrystallization occurring during FSW of aluminum alloys, e.g., [37]. It explains the substantial material hardening revealed in the stir zone (Figure 9).

## 4. Summary

The present work was undertaken in order to tailor the dissimilar FSW of aluminum and titanium alloys in the lap-welding configuration. In contrast to the previous studies in this area, the proposed approach was based on two principal issues: (i) the zero penetration depth of the welding tool into the titanium part and (ii) the high FSW heat input. From experimental observations, it was shown that the “zero-penetration-depth” approach is feasible for dissimilar lap welding of aluminum and titanium alloys, if FSW heat input is sufficiently high. Specifically, sound welds were produced using an ordinary cost-effective tool, with no measurable tool abrasion and no dispersion of harmful titanium fragments within the aluminum side. Moreover, due to the lack of direct contact between the welding tool and titanium side, the intermetallic layer was as narrow as ~0.1 µm. This provided excellent bond strength between aluminum and titanium, so the welded joints typically failed in the aluminum part. In other words, the intermetallic compound was not a critical structural element, in contrast to the typical dissimilar FSW joints. Instead, the mechanical performance of the dissimilar welds was typically governed by the strength of the aluminum part.

On the other hand, several important limitations of the approach were revealed. First and foremost, the relatively high-heat input promoted significant microstructural changes in the aluminum part. In the work-hardened material condition used in the present study, pronounced recrystallization and grain growth were observed. This resulted in the joint efficiency being as low as 60–70%.

An essential issue is also a comparatively narrow processing window. Specifically, the approach is only applicable at high welding temperatures and low welding speeds. The additional requirement is also a relatively small distance between the probe tip and the interphase surface (≈50 µm in the present study). This imposes strict limitations on the accuracy of FSW machines and operator skills.

Moreover, the proposed approach may not be feasible for highly alloyed aluminum alloys. The excessively high impurity content may retard the diffusion mobility of aluminum atoms and, thus, restrict their penetration into the titanium side.

## Figures and Tables

**Figure 1 materials-15-08418-f001:**
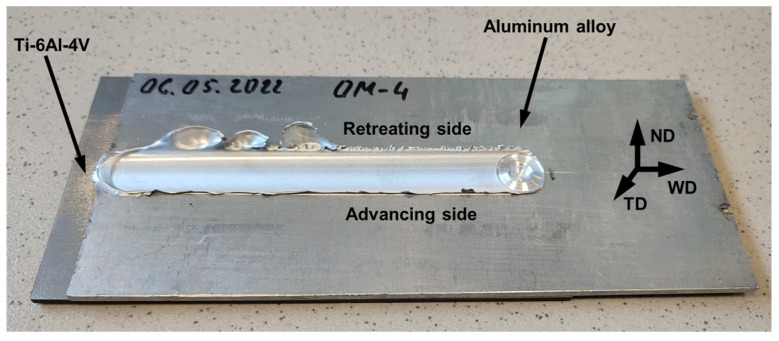
Typical appearance of welded joints. ND, WD, and TD are normal direction, welding direction, and transverse direction, respectively. Note: The photo was taken from the welded joint of AA6013 alloy produced at a welding speed of 50.8 mm/min.

**Figure 2 materials-15-08418-f002:**
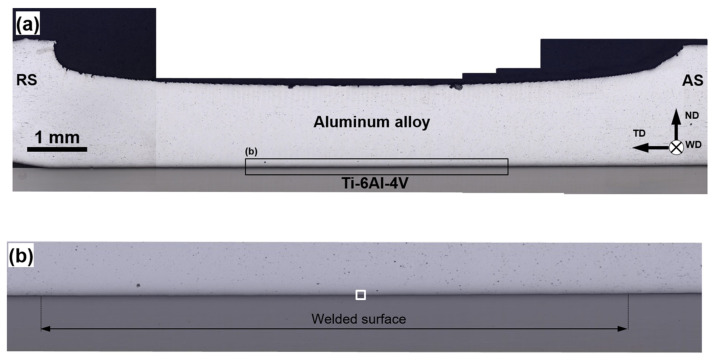
Typical cross-sectional optical image of welded joints (unetched condition): (**a**) low-magnification overview, (**b**) welded surface. ND, TD, and WD are normal direction, transverse direction, and welding direction, respectively. RS and AS are retreating side and advancing side, respectively. Note: the selected area in (**b**) shows the typical position of SEM and EDS observations. Note 2: The optical micrographs were taken from the welded joint of AA6013 alloy produced at a welding speed of 76.2 mm/min.

**Figure 3 materials-15-08418-f003:**
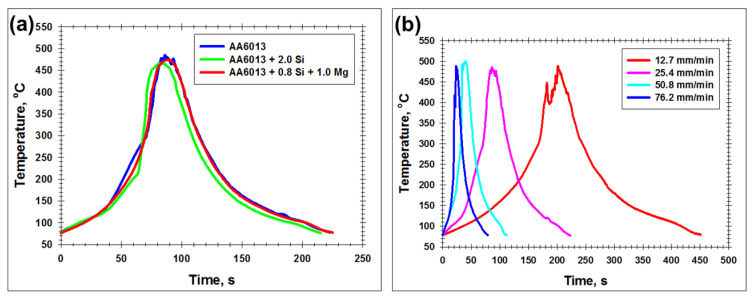
Typical effect of the alloying composition of aluminum alloy (**a**) and welding speed (**b**) on FSW thermal cycle. In (**a**), FSW was conducted at welding speed of 25.4 mm/min. In (**b**), temperature data for AA6013 alloy are shown.

**Figure 4 materials-15-08418-f004:**
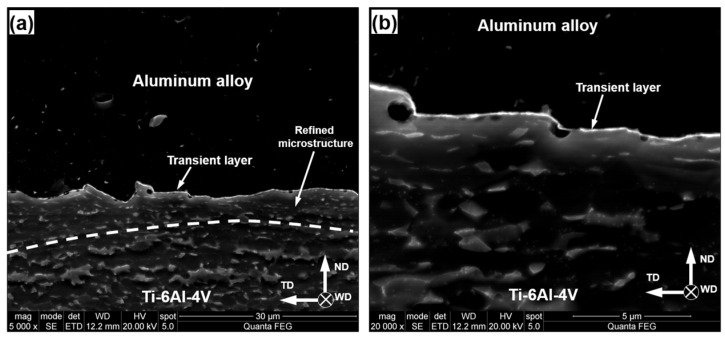
Typical SEM micrographs of welded surface (etched condition): (**a**) low-magnification image, (**b**) high-magnification image. ND, TD, and WD are normal direction, transverse direction, and welding direction, respectively. Note: the beta phase in Ti–6Al–4V appears bright. Note 2: The SEM micrographs were taken from the welded joint of AA6013 alloy produced at the welding speed of 76.2 mm/min.

**Figure 5 materials-15-08418-f005:**
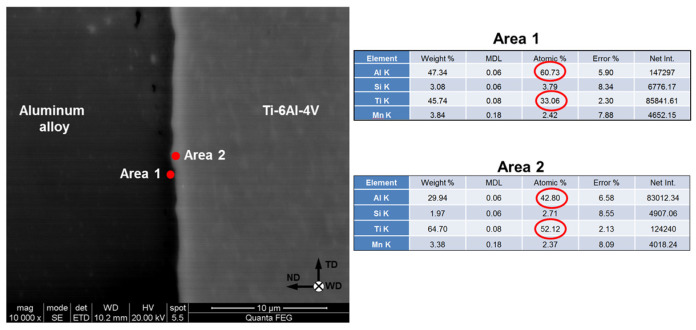
Typical results of quantitative EDS analysis of intermetallic layer. ND, TD, and WD are normal direction, transverse direction, and welding direction, respectively. Note: The EDS data were taken from the welded joint AA6013 alloy produced at the welding speed of 12.7 mm/min.

**Figure 6 materials-15-08418-f006:**
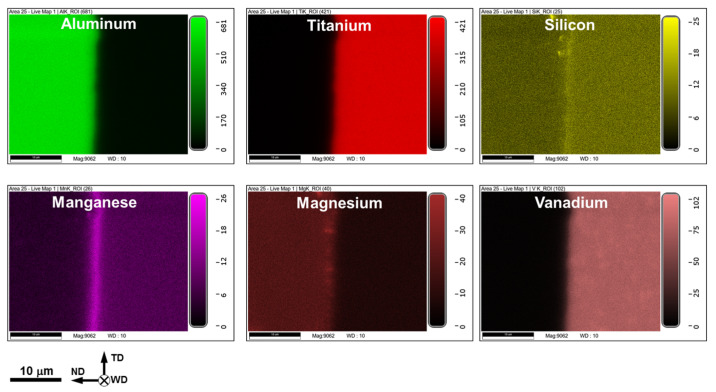
Typical EDS maps showing elemental distribution across the welded surface. In all cases, aluminum side is on the left and Ti–6Al–4V side is on the right. ND, TD, and WD are normal direction, transverse direction, and welding direction, respectively. Note: The EDS data were taken from the welded joint of AA6013 alloy produced at the welding speed of 76.2 mm/min.

**Figure 7 materials-15-08418-f007:**
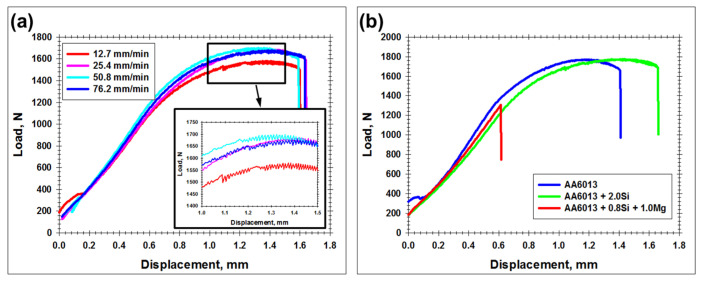
Typical effect of welding speed (**a**) and impurity content in aluminum alloy (**b**) on tensile behavior of welded joints. In (**a**), tensile diagrams for AA6013 + 2.0 Si alloy are shown. In (**b**), welded joints were produced at welding rate of 76.2 mm/min. Note: The high-magnification insert in the bottom-right corner of (**a**) illustrates Portevin–Le Chatelier effect.

**Figure 8 materials-15-08418-f008:**
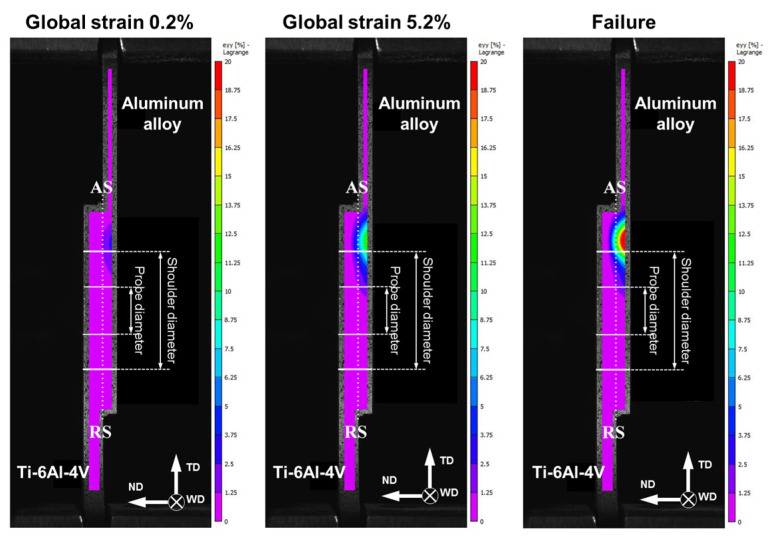
Typical distribution of longitudinal strains measured by digital image correlation technique in transverse cross section during lap-shear tests of welded joints as a function of global strain. Tensile direction is vertical. ND, TD, and WD are normal direction, transverse direction, and welding direction, respectively. RS and AS are retreating side and advancing side, respectively. Note: The tensile data were taken from the welded joint of AA6013 alloy produced at a welding speed of 12.7 mm/min.

**Figure 9 materials-15-08418-f009:**
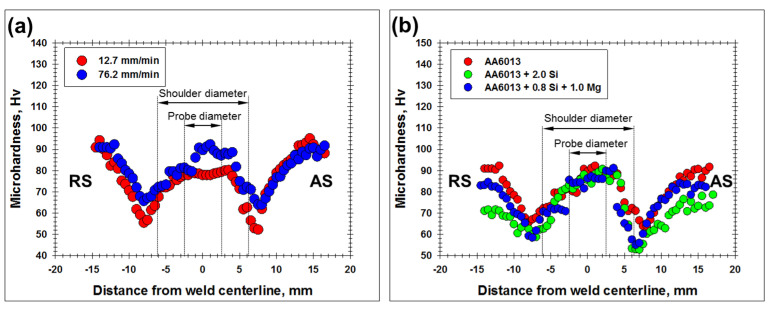
Typical effect of welding speed (**a**) and impurity content in aluminum alloy (**b**) on microhardness profile measured across the mid-thickness of aluminum part of dissimilar joint. In (**a**), microhardness data for AA6013 alloy are shown. In (**b**), welded joints were produced at welding rate of 76.2 mm/min.

**Figure 10 materials-15-08418-f010:**
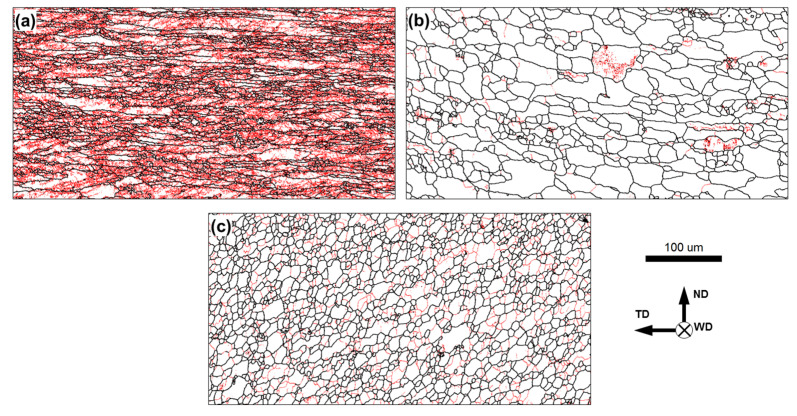
Typical EBSD grain boundary maps showing microstructure in (**a**) initial material, (**b**) heat-affected zone, and (**c**) stir zone. In the maps, low-angle boundaries (LABs) and high-angle boundaries (HABs) are depicted as red and black lines, respectively. The scale bar and reference frame for all maps are shown in the bottom-right corner. Note: EBSD data were taken from the welded joint of AA6013 alloy produced at the welding speed of 12.7 mm/min.

**Table 1 materials-15-08418-t001:** The nominal chemical composition of commercial aluminum alloy 6013 (wt.%).

Al	Mg	Si	Cu	Mn	Fe	Zn	Cr	Ti
Bal.	1.0	0.8	0.8	0.5	0.5	0.25	0.1	0.1

**Table 2 materials-15-08418-t002:** The experimental aluminum alloys used in the present work.

AA6013 + 2.0 Si	AA6013 + 0.8 Si + 1.0 Mg	AA6013 + 3.0 Mg	AA6013 + 5.0 Mg

Note: The extra-alloying content is given in wt.%.

**Table 3 materials-15-08418-t003:** Effect of chemical composition of aluminum alloys and welding speed on cross-sectional width of welded surface.

Aluminum Alloy	AA6013		AA6013 + 2.0 Si		AA6013 + 0.8 Si + 1.0 Mg	
Welding speed, mm/min	12.7	76.2	12.7	76.2	12.7	76.2
Width of welded surface, mm	6.575	5.875	5.123	4.115	5.106	3.138

## Data Availability

The data presented in this study are available on request from the corresponding author.

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
