# Peer review of "Tailoring of Dissimilar Friction Stir Lap Welding of Aluminum and Titanium"

_materials, 2022, doi:10.3390/ma15238418_

Round 1
Reviewer 1 Report
The Paper titled “Tailoring of dissimilar friction-stir lap welding of aluminum and titanium” As long as the work's assumptions are correct and could constitute an interesting contribution to the discipline, and analyses properties allow me to recommend the article for publication. The following are only some of the comments that influenced my decision, which recommends that the manuscript be applied before its resubmission.
Absence of Novelty-The manuscript as it looks like similar works and observations were reported by other researchers.
In fig 4 mark all the parts such as TMAZ, HAZ etc. & include EDX graph for better understanding of the beginners.
The author includes experimental trial specimens and include standard deviation for the graph (Fig 7)
The author include experimental setup for fabrication.
Its better to replace the current keywords by some special keywords
The author should depict the flow graph to illustrate the need of the proposed.
Approach in Comparison with recent study and methods would be appreciated.
The author include intermetallic compound formation occurs at the weld interface.
Author Response
Responses to the comments of Reviewer #1
on the paper entitled “Tailoring of dissimilar friction-stir lap welding of aluminum and titanium” (Manuscript ID: materials-2000303)
The authors would like to express their gratitude to Reviewer for his/her remarks. Below, we provided specific replies to the issues raised.
The Paper titled “Tailoring of dissimilar friction-stir lap welding of aluminum and titanium” As long as the work's assumptions are correct and could constitute an interesting contribution to the discipline, and analyses properties allow me to recommend the article for publication. The following are only some of the comments that influenced my decision, which recommends that the manuscript be applied before its resubmission.
Absence of Novelty-The manuscript as it looks like similar works and observations were reported by other researchers.
Authors’ response
According to the comment, the novelty of this manuscript has been emphasized in Introduction and Summary sections of the revised manuscript.
Introduction (Page 2, Paragraph 7):
“Therefore, the present work was undertaken in an attempt to tailor the dissimilar FSW of aluminum and titanium alloys in the lap-welding configuration. In contrast to the previous works in this area, the proposed approach was based on two principal issues: (i) the zero penetration depth of the welding tool into the titanium part and (ii) the high FSW heat input.”
Summary (Page 10, Paragraph 1):
“The present work was undertaken in order to tailor the dissimilar FSW of aluminum and titanium alloys in the lap-welding configuration. In contrast to the previous studies in this area, the proposed approach was based on two principal issues: (i) the zero penetration depth of the welding tool into the titanium part and (ii) the high FSW heat input. From experimental observations, it was shown that the “zero-penetration-depth” approach is feasible for dissimilar lap welding of aluminum and titanium alloys, if FSW heat input is sufficiently high. Specifically, sound welds were produced using an ordinary cost-effective tool, with no measurable tool abrasion and no dispersion of harmful titanium fragments within the aluminum side. Moreover, due to the lack of direct contact between the welding tool and titanium side, the intermetallic layer was as narrow as ~0.1 m. This provided excellent bond strength between aluminum and titanium, so the welded joints typically failed in the aluminum part. In other words, the intermetallic compound was not a critical structural element, in contrast to the typical dissimilar FSW joints. Instead, the mechanical performance of the dissimilar welds was typically governed by the strength of the aluminum part.”
In fig 4 mark all the parts such as TMAZ, HAZ etc. & include EDX graph for better understanding of the beginners.
Authors’ response
The EDX data were taken from central part of the welded surface. This microstructural region represented the bottom side of the stir zone in aluminum part. To avoid misunderstanding, appropriate remark has been added to the revised manuscript (Page 5, Section 3.3, Paragraph 1):
“In order to get insight into the welding mechanism, the interphase surface between aluminum and titanium parts was studied using SEM and EDS techniques. In all cases, observations were conducted at the center of the welded surface, as indicated in Fig. 2b.”
Moreover, the typical location of EDS measurements has been indicated in Fig. 2b in the revised manuscript.
The author includes experimental trial specimens and include standard deviation for the graph (Fig 7)
Authors’ response
According to the comment, the sketch of the specimens used for the lap-shear test has been shown in the revised manuscript (as supplementary Fig. S3).
In order to illustrate experimental scattering during lap-shear tests, the entire set of the recorded deformation diagrams has been added to the revised manuscript (as supplementary Fig. S4).
Page 7, Section 3.4, Paragraph 1:
“The typical results of lap-shear tests are shown in Figs. 7. The entire set of the deformation diagrams is summarized in supplementary Fig. S4.”
The author include experimental setup for fabrication.
Authors’ response
According to the comment, the experiment setup has been added to the revised manuscript (as supplementary Fig. S1):
Its better to replace the current keywords by some special keywords
Authors’ response
According to the comment, the special keywords have been provided to the revised manuscript:
“Keywords: Dissimilar friction-stir lap welding; 6013 aluminum alloy; Ti-6Al-4V alloy; Intermetallic compound; Grain structure; Lap-shear test”
The author should depict the flow graph to illustrate the need of the proposed
Authors’ response
According to the comment, the main idea of this work has been arranged as Graphical abstract, which has been added to the revised manuscript.
Approach in Comparison with recent study and methods would be appreciated.
Authors’ response
According to the comment, the proposed approach has been discussed in the context of the previous works in this area (Section 1, Page 2, Paragraph 7):
“Therefore, the present work was undertaken in an attempt to tailor the dissimilar FSW of aluminum and titanium alloys in the lap-welding configuration. In contrast to the previous works in this area, the proposed approach was based on two principal issues: (i) the zero penetration depth of the welding tool into the titanium part and (ii) the high FSW heat input.”
The author include intermetallic compound formation occurs at the weld interface.
Authors’ response
According to the comment, the formation of intermetallic compounds has been discussed in the revised manuscript (Section 1, Pages 1 to 2)
“The formation of intermetallic compounds at the aluminum-titanium interface is also of great concern [1-3]. Usually, TiAl3 intermetallic has been reported to form [2, 3, 5-19], presumably due to the comparatively-low free energy of intermetallic reaction in this case [20]. The details of this process are unclear, however. In some cases, TiAl [3, 6, 7, 9, 10, 15] and Ti3Al [2, 3, 6, 21] phases have also been found. As shown by Choi et al. [6], the development of TiAl and Ti3Al intermetallics is associated with a diffusion penetration of aluminum through the early-formed TiAl3 phase. Hence, the formation of TiAl and Ti3Al compounds occurs sequentially, one by one; moreover, the probability of these two processes is directly linked to FSW heat input. Specifically, it was shown that the increase in the tool rotation rate promotes the nucleation of the TiAl phase first and then the Ti3Al phase [6].”

Reviewer 2 Report
This paper reports an approach to optimize dissimilar friction stir lap welding of aluminum and titanium alloys. The method has a good weld seam, no tool wear and no harmful titanium fragments scattered on the aluminum side, with good bond strength. However, there is also the limitation that the microstructure of the aluminum part is affected and cannot be applied to high aluminum alloy. It can provide a reference for related research. However, some of the conclusions of the manuscript need sufficient evidence and there are some detailed aspects that need further modification.
(Introduction)
It is suggested to clarify the difference between TiAl and Ti3Al when forming.
(Materials and Methods)
It is suggested to display the composition content of AA6013 aluminum alloy in the table to explain the reason for increasing Mg and Si (for example, Mg and Si are important constituent elements in aluminum alloy).
(Results and Discussion)
" No obvious titanium fragments are found in aluminum alloy ", it is recommended to provide microscope pictures or SEM-EDS mapping scans or other evidence to support this argument..
How to get “0.4” in “this temperature represented only ≈ 0.4 of the homologous temperature of Ti-6Al-4V”. If it is obtained from previous research, relevant articles should be pointed out.
It is suggested that the refinement process of beta phase particles shown in Figure 4 should be compared with that without refinement, to prove that plastic deformation occurs at the upper part of Ti6Al4V side.And the specific position of beta phase particles in the figure shall be marked.
“no clear correlation between FSW conditions and the thickness or chemical composition of intermetallic compounds was found.”, is it concluded that the thickness of intermetallic compounds obtained under different FSW conditions has been counted? There is no picture in the article to explain this point, why not show it?
(Summary)
In the summary of "the intermediate layer was as narrow as~0.1 µm.", this view was not shown in the part of Results and Discussion. This view is directly obtained without material characterization as a basis.
Author Response
Responses to the comments of Reviewer #2
on the paper entitled “Tailoring of dissimilar friction-stir lap welding of aluminum and titanium” (Manuscript ID: materials-2000303)
The authors would like to express their gratitude to Reviewer for his/her remarks. Below, we provided specific replies to the issues raised.
This paper reports an approach to optimize dissimilar friction stir lap welding of aluminum and titanium alloys. The method has a good weld seam, no tool wear and no harmful titanium fragments scattered on the aluminum side, with good bond strength. However, there is also the limitation that the microstructure of the aluminum part is affected and cannot be applied to high aluminum alloy. It can provide a reference for related research. However, some of the conclusions of the manuscript need sufficient evidence and there are some detailed aspects that need further modification.
(Introduction)
It is suggested to clarify the difference between TiAl and Ti3Al when forming.
Authors’ response
According to the comment, the recommended issue has been discussed in the revised manuscript (Section 1, Pages 1 to 2):
“The formation of intermetallic compounds at the aluminum-titanium interface is also of great concern [1-3]. Usually, TiAl3 intermetallic has been reported to form [2, 3, 5-19], presumably due to the comparatively-low free energy of intermetallic reaction in this case [20]. The details of this process are unclear, however. In some cases, TiAl [3, 6, 7, 9, 10, 15] and Ti3Al [2, 3, 6, 21] phases have also been found. As shown by Choi et al. [6], the development of TiAl and Ti3Al intermetallics is associated with a diffusion penetration of aluminum through the early-formed TiAl3 phase. Hence, the formation of TiAl and Ti3Al compounds occurs sequentially, one by one; moreover, the probability of these two processes is directly linked to FSW heat input. Specifically, it was shown that the increase in the tool rotation rate promotes the nucleation of the TiAl phase first and then the Ti3Al phase [6].”
(Materials and Methods)
It is suggested to display the composition content of AA6013 aluminum alloy in the table to explain the reason for increasing Mg and Si (for example, Mg and Si are important constituent elements in aluminum alloy).
Authors’ response
The manuscript has been revised according to the comment (Section 2, Pages 2 to 3):
“The baseline materials used in the present study included commercial AA6013 aluminum alloy and Ti-6Al-4V titanium alloy. The nominal chemical composition of AA6013 aluminum alloy is shown in Table 1. In order to investigate the possible influence of the important alloying elements in aluminum alloys (i.e., magnesium and silicon) on the feasibility of FSW, four additional experimental aluminum alloys were also utilized, as shown in Table 2. All aluminum alloys were produced by semi-continuous casting using laboratory equipment at Belgorod National Research University. The cast ingots were homogenized at 550oC for 4 hours and then cold-rolled to a final thickness of 2 mm (≈80% of total thickness reduction). Titanium alloy was received as 2-mm thick plates in the mill-annealed condition.”
(Results and Discussion)
" No obvious titanium fragments are found in aluminum alloy ", it is recommended to provide microscope pictures or SEM-EDS mapping scans or other evidence to support this argument.
Authors’ response
According to the comment, appropriate SEM-EDS maps have been provided in the revised manuscript (Fig. 6):
How to get “0.4” in “this temperature represented only ≈ 0.4 of the homologous temperature of Ti-6Al-4V”. If it is obtained from previous research, relevant articles should be pointed out.
Authors’ response
According to the comment, this issue has been discussed in the revised manuscript (Page 5, Section 3.2, Paragraph 2):
“Notably, given the liquidus temperature of Ti-6Al-4 of 1660oC (or 1933 K), the peak FSW temperature (i.e., 773 K) represented only 773/1933≈0.4 of the homologous temperature of titanium alloy, thus being relatively low. Accordingly, no essential diffusion activity of titanium could be expected.”
It is suggested that the refinement process of beta phase particles shown in Figure 4 should be compared with that without refinement, to prove that plastic deformation occurs at the upper part of Ti6Al4V side. And the specific position of beta phase particles in the figure shall be marked.
Authors’ response
Figure 4 has been revised according to the comment:
“no clear correlation between FSW conditions and the thickness or chemical composition of intermetallic compounds was found.”, is it concluded that the thickness of intermetallic compounds obtained under different FSW conditions has been counted? There is no picture in the article to explain this point, why not show it?
Authors’ response
Yes, the thickness and chemical composition of intermetallic layers have been checked systematically for all studied FSW conditions. To avoid misunderstanding, this issue has been emphasized in the revised manuscript (Page 7, Paragraph 4):
“Surprisingly, no clear correlation between FSW conditions and the thickness or chemical composition of intermetallic compounds was found. In all FSW conditions examined in the present study, the intermetallic compound included TiAl3 and TiAl phases and the total thickness of the intermetallic layer was of ~0.1 micron, as exemplified in Figs. 5 and 4, respectively.”
(Summary)
In the summary of "the intermediate layer was as narrow as~0.1 µm.", this view was not shown in the part of Results and Discussion. This view is directly obtained without material characterization as a basis.
Authors’ response
In the revised manuscript, the thickness of intermetallic layer has been shown in Fig. 4 and discussed in Section 3.3. (Pages 5 to 6):
“In all FSW conditions studied in the present work, SEM observations revealed the formation of a thin (~0.1 micron) transition layer between aluminum alloys and Ti-6Al-4V. A typical example is indicated by arrow in Fig. 4a.”
